# Virulence Mechanisms of Staphylococcal Animal Pathogens

**DOI:** 10.3390/ijms241914587

**Published:** 2023-09-26

**Authors:** Gordon Y. C. Cheung, Michael Otto

**Affiliations:** Pathogen Molecular Genetics Section, Laboratory of Bacteriology, National Institute of Allergy and Infectious Diseases, National Institutes of Health, 50 South Drive, Bethesda, MD 20814, USA; cheunggo@niaid.nih.gov

**Keywords:** *Staphylococcus aureus*, MRSA, mastitis, *Staphylococcus pseudintermedius*, *Staphylococcus hyicus*, *Staphylococcus chromogenes*, *Staphylococcus agnetis*, *Staphylococcus xylosus*

## Abstract

Staphylococci are major causes of infections in mammals. Mammals are colonized by diverse staphylococcal species, often with moderate to strong host specificity, and colonization is a common source of infection. Staphylococcal infections of animals not only are of major importance for animal well-being but have considerable economic consequences, such as in the case of staphylococcal mastitis, which costs billions of dollars annually. Furthermore, pet animals can be temporary carriers of strains infectious to humans. Moreover, antimicrobial resistance is a great concern in livestock infections, as there is considerable antibiotic overuse, and resistant strains can be transferred to humans. With the number of working antibiotics continuously becoming smaller due to the concomitant spread of resistant strains, alternative approaches, such as anti-virulence, are increasingly being investigated to treat staphylococcal infections. For this, understanding the virulence mechanisms of animal staphylococcal pathogens is crucial. While many virulence factors have similar functions in humans as animals, there are increasingly frequent reports of host-specific virulence factors and mechanisms. Furthermore, we are only beginning to understand virulence mechanisms in animal-specific staphylococcal pathogens. This review gives an overview of animal infections caused by staphylococci and our knowledge about the virulence mechanisms involved.

## 1. Introduction

Many staphylococci, above all *Staphylococcus aureus*, are important human pathogens, causing many moderately severe skin and soft tissue infections, but also severe and often fatal infections such as sepsis, osteomyelitis, endocarditis, and pneumonia [1]. In animals, staphylococci can cause similar infections, which in addition to animal welfare issues result in a massive financial cost for animal farmers. Arguably the most notorious and overall costly type of animal infection that is predominantly due to staphylococci is mastitis in cows, which in 2011 was estimated to cost the global dairy industry $20–$33 billion per year [2]. In addition to the economic burden, staphylococcal infections in livestock represent a source of transfer of antimicrobial resistance to humans. There is a massive use of antibiotics for such infections, which results in widespread occurrence of resistant staphylococcal strains, such as livestock-associated methicillin-resistant *S. aureus* (LA-MRSA) [3].

Staphylococci, grouped in the genus *Staphylococcus* with more than 80 species, are non-motile facultative anaerobic Gram-positive cocci [4]. All species are part of the natural epithelial microbiota of mammals. Many, such as *S. aureus* or *S. epidermidis*, are found in a variety of hosts, while other species show pronounced host specificity (e.g., *S. hyicus* in swine [5], *S. pseudintermedius* in dogs [6], or *S. felis* in cats [7]). However, specific lineages of a species, as has been shown for *S. aureus*, may be adapted to a specific host [8]. Host specificity does not seem to be absolute. *S. hyicus*, for example, can also be found in other mammals [9,10]. Based on their ability to produce coagulase, staphylococci are divided into the commonly more pathogenic coagulase-positive staphylococci (CoPS), with *S. aureus* as the major species, and the less pathogenic coagulase-negative staphylococci (CoNS), such as *S. epidermidis* [4].

From a veterinary perspective, the most important pathogens are *S. aureus* and other CoPS, namely *S. intermedius*, *S. pseudintermedius*, *S. delphini*, *S. hyicus*, and *S. cornubiensis*, which are members of the *S. intermedius* group, as well as *S. lutrae*, *S. agnetis*, and *S. schleiferi* [11]. Of these species, some can show variability in coagulase positivity. Based on whole-genome sequencing, it has recently been proposed that the two subspecies of *S. schleiferi* be classified as their own distinct species (*S. schleiferi and S. coagulans*) [12].

CoNS, particularly *S. epidermidis*, have more recently drawn much attention due to potential beneficial functions in the human microbiome by contributing to skin barrier homeostasis [13] and fighting pathogens directly or after engaging the host’s immune system [14,15,16]. However, traditionally they are known for their capacity to cause biofilm-associated medical device infections and concomitant complications, such as septicemia, in humans [17,18]. As animals only rarely undergo similar medical interventions, the importance of CoNS as animal pathogens is generally low. Nevertheless, some CoNS have been reported to cause infections in animals, such as *S. felis* [19]. These infections may occur only in an opportunistic fashion in compromised hosts, or remain subclinical, such as subclinical mastitis caused by CoNS [20]. Furthermore, as is often observed in humans, attribution of clinical symptoms to CoNS may be due to sample contamination rather than a true pathogenic role [21]. However, CoNS may represent an important reservoir for the transfer of antimicrobial resistance to CoPS of veterinary importance [22,23].

In humans, based on research with *S. aureus*, it is well established that staphylococcal infections originate from asymptomatic colonization or contaminated fomites [24,25]. One can assume that the same is true for animal infections, although evidence is scarcer. There is, however, some evidence underscoring that notion, indicating that *S. aureus* isolates from mammary and extra-mammary sites, such as particularly hock skin, are genetically related [26].

Research into virulence factors and mechanisms of the staphylococci has mostly focused on *S. aureus* and, among the CoNS, *S. epidermidis* [1,27]. Generally, *S. aureus* and, to a more limited degree, other CoPS have a large arsenal of virulence factors of different sorts, comprising many toxins, adhesion molecules, immune evasion factors, as well as a complicated network of regulatory mechanisms to control their production [1]. CoNS, in contrast, have much fewer virulence factors, which also tend to be more passive in nature [28].

In this review, we will first give an overview of staphylococcal virulence factors and antimicrobial resistance. We then present the main species of animals in which staphylococcal infections have been observed and investigated, provide an overview of the major infection types and staphylococcal species involved, and present and discuss main underlying virulence mechanisms.

## 2. Overview of Staphylococcal Virulence Factors

All virulence mechanisms of staphylococci aim to increase survival after the bacteria have breached the epithelial barrier of the skin or mucosal surfaces, where they reside in an asymptomatic fashion, or to achieve that breach. Although staphylococcal asymptomatic colonization is an important source of infection [24], factors that only facilitate asymptomatic survival on the epithelia are not considered virulence factors.

Virulence factors facilitating the transition from colonization to systemic infection mostly comprise alpha-toxin [29], which facilitates invasion through the keratinocyte layer, and other toxins that promote skin abscess formation, such as several leukocidins (leukotoxins) and phenol-soluble modulins (PSMs) [30,31].

Once in the bloodstream, the bacteria are attacked by manifold modes of immune defense mechanisms. *S. aureus* and, to a more limited extent, other staphylococci, have a plethora of immune evasion factors to counter host immune defenses [32,33]. These can be categorized into passive defense mechanism, such as capsule, exopolysaccharide, or biofilm formation [34,35,36], mechanisms that alter the nature of bacterial structures by which the immune system recognizes the invader (for example, enzymes that remove formyl-methionine from proteins or fatty acids from lipoproteins [37,38]), and molecules that block specific steps of innate immune defense mechanisms, e.g., within the complement enzymatic cascade [39]. Furthermore, surface proteins, many of which belong to the family called “microbial surface components recognizing adhesive matrix molecules” (MSCRAMMs), facilitate adhesion to host tissues, and some are involved in abscess formation in organs [40,41,42]. Examples include the collagen-binding protein Cna and the fibronectin-binding proteins FnBPA and FnBPB. On the other hand, especially *S. aureus* produces a series of aggressive toxins that can directly eliminate immune cells. The most important among those are the abovementioned alpha-toxin, PSMs, and the family of leukocidins [30,43,44]. Alpha-toxin is a pore-forming toxin that forms a heptameric pore in target cells after binding to the ADAM10 receptor [44]. PSMs are detergent-like peptides that have membrane-perturbing cytolytic capacity toward a variety of cell types in a non-receptor-dependent fashion [45]. As for the leukocidin family, five different leukocidins exist in *S. aureus* that are associated with human infections: leukocidin ED (LukED), Panton–Valentine leukocidin (PVL or LukSF–PV), gamma-hemolysins AB and CB (HlgAB and HlgCB), and leukocidin AB (LukAB; also known as LukGH) [43]. LukF′M [46], LukPQ [47], and LukI [48] are associated with bovine, equine, and canine infections, respectively. Leukocidin functionality requires S and F subunits, in which the S-component first recognizes host-specific cell immune receptors with high-affinity [49], followed by the recruitment of the F-component and the subsequent assembly of an octameric beta-barrel pore into the host plasma membrane lipid bilayer. Notably, leukocidins can have pronounced and alpha-toxin has limited target species specificity [44,49]. In contrast, owing to their mechanism, PSMs do not show such specificity [45]. Exfoliating toxins are another class of toxins that are produced by *S. aureus* and certain other staphylococcal species. To date, five members (ETA, ETB, ETC, ETD, and ETE) have been described in *S. aureus* [50,51,52,53], and, according to the presence of a triad of conserved catalytic residues [54], are categorized as glutamate-specific serine proteases [51]. The exfoliative toxins specifically target the cleavage of a single peptide bond within the extracellular domain of desmoglein-1 (Dsg-1) [55,56], which is expressed ubiquitously in stratified squamous epithelia [57] and is involved in intercellular adhesion [58]. ETA, ETB, and ETD are found in human isolates, whereas ETC and ETE were discovered in horse and ovine isolates, respectively [50,51,52,53].

While the control of staphylococcal infections is predominantly due to innate mechanisms of host defense, *S. aureus* also targets acquired (antibody-based) immunity by producing protein A, a protein that binds to the invariant Fc part of IgG molecules, thus producing what has been called a “camouflage coat” of non-specific antibodies on the bacterial surface [59]. Additionally, protein A skews the immune response away from recognizing other virulence factors by eliciting the production of B cells that almost exclusively recognize protein A [60].

Further staphylococcal virulence factors comprise a superfamily of over 20 superantigens [61], which lead to an overshooting immune response with pronounced cytokine secretion by activating T cells in a non-specific manner [62], proteases, which degrade host tissue for nutrient acquisition but also have more specific roles in destroying specific immune factors, such as complement factors and antimicrobial peptides [39,63], and many further proteins, such as coagulase, the enzyme on which the CoNS–CoPS classification is based. Coagulase converts fibrinogen into fibrin, thereby causing blood plasma to clot [64].

For more detail, the reader is referred to reviews that have described the virulence factors of *S. aureus*, many of which we have cited above. Here, it shall only be noted in conclusion that virulence mechanisms of *S. aureus* are subject to strict control by a plethora of regulatory systems [1,65]. Among them, the quorum-sensing virulence regulator Agr is probably the most important and best described [66]. Agr positively controls virtually all toxins, degradative exoenzymes, and similar virulence factors, with having an exceptionally direct and strict control of PSM production [67]. This is believed to postpone the production of aggressive virulence factors, many of which directly or indirectly stimulate host defense, until those host defenses can be countered by a sufficiently large infectious bacterial population. It is generally believed that Agr controls all surface proteins in a negative fashion (to limit their production to the beginning of an infection when adherence to host tissues is the most important task) [68], but more recent research in clinically important *S. aureus* has shown that this is true only for some surface proteins, such as protein A [69]. Notably, there are frequently different subgroups of Agr in a species, and the auto-inducing peptide (AIP) Agr extracellular signal of a non-self species or subgroup often is inhibitory to Agr by competitive interaction at the AgrC histidine kinase AIP receptor [66,70].

CoPS other than *S. aureus* have the potential to produce several of those virulence factors. We will present this in the dedicated chapters of this review. In contrast, CoNS only very rarely produce toxins other than PSMs and generally possess only passive virulence mechanisms such as biofilm formation [28]. The virulence factors of the CoNS are believed to have an original role in the asymptomatic commensal state, some of which are of additional value during infection owing to their intrinsic characteristics but—unlike *S. aureus* virulence factors—do not seem to be produced specifically to promote infection [27].

## 3. Antimicrobial Resistance

Staphylococcal infections are often difficult to treat due to antimicrobial resistance [71]. Many staphylococci have a pronounced capacity to form biofilms, which provide largely non-specific resistance (or more correctly, tolerance) to virtually all antibiotics [34]. Biofilm formation is involved in device-associated infection, but also many other infections, such as endocarditis. Despite intensive efforts, no efficient drugs targeting biofilms have been developed; as a result, biofilms still represent an enormous problem in the clinic [72].

However, when talking about antimicrobial resistance, most people mean resistance that is due to specific resistance genes and targets specific antibiotics. This type of resistance is due to many different mechanisms, such as drug export, change of target structures, or degradation [73]. The more a particular antibiotic is used, the more likely it is that resistant strains develop and spread. Furthermore, resistance genes can often be transferred between strains and species by horizontal gene transfer [74].

Staphylococcal resistance to most antibiotics in use has been reported in animals [75,76]. The frequency of antibiotic-resistant strains in animal infections often exceeds that found in humans, because antibiotics are used much more deliberately in animal agriculture, not only to prevent infections but also to increase growth [76,77]. In the United States, about 80% of all used antibiotics are used in animal agriculture, and of those 70% are deemed “medically important” (for humans) [78]. In many countries, for example in Asia, antibiotic use in livestock considerably exceeds that in the US or Europe [79].

Resistance to methicillin, afforded by the presence of the *mec*A gene in a mobile genetic element (MGE) called staphylococcal cassette chromosome *mec* (SCC*mec*), is the most important antibiotic resistance determinant in staphylococci infecting animals. The *mec*A gene codes for a transpeptidase (penicillin-binding protein 2A [PBP2A]), which has a critical role in the synthesis of the bacterial cell wall [80]. Strains that have acquired the *mec*A gene are able to grow in the presence of beta-lactam antibiotics because PBP2A has low affinity for these drugs. In contrast, strains harboring native PBPs are unable to grow under the same conditions due to their greater affinity for beta-lactams [81].

Livestock-associated MRSA (LA-MRSA) is an enormous problem in animal farming [82]. Despite considerable reduction in antibiotic use, for example in some European countries, the frequency of LA-MRSA has barely declined [83]. This may be explained by the only minor fitness cost that methicillin resistance genes impose on the bacterial host, especially in some MRSA lineages, meaning that, once specific types of MRSA have spread, it is difficult to reduce them in the animal (or human) population [84]. It has been shown that humans in close contact with livestock, such as pig farmers or veterinarians, are at increased risk of being colonized and infected with LA-MRSA [85,86]. However, likely since human MRSA and LA-MRSA comprise different lineages with different host adaptation characteristics, LA-MRSA has not been a considerable factor that has contributed to the human CA-MRSA epidemic of the last two decades [82,87], although LA-MRSA human infections may be severe and fatal when they occur [88]. Finally, pet animals may be responsible for recurring infections in households, despite MRSA colonization of pet animals being deemed transient [89].

## 4. Staphylococcal Infections in Animal Hosts

In this section, we will present staphylococcal infections in animals, sorted according to the animal host. An overview of main infection types and staphylococcal species typically causing infections in a specific host is given in Table 1. Main virulence factors associated with a specific staphylococcal species and demonstrated or assumed function in animal infections are shown in Table 2.

### 4.1. Staphylococcus Infections in Ruminants (Cattle, Sheep, and Goats)

While staphylococci are likely able to generally cause similar systemic and, to a certain extent, skin infections in ruminants as in humans, those that are most important from a clinical and agricultural perspective are infections of the udder, i.e., mastitis. In cattle, goats, and sheep, staphylococci can cause intramammary infections (IMIs): subclinical or, less frequently, clinical mastitis, with the latter being distinguished from subclinical mastitis by visible abnormalities in the milk and swelling or tenderness of the udder, with a frequent presence of pus [117]. In severe cases, usually due to *S. aureus*, it can develop into necrotizing gangrene [118]. CoNS, including *S. chromogenes*, *S. simulans*, *S. haemolyticus*, *S. xylosus*, *S. epidermidis*, and several other species, are the most frequent causes of subclinical mastitis [20,95,100].

Ruminants show similar *S. aureus* carriage rates as humans, ~20–30% [119], and many non-*S. aureus* species are assumed to be widespread in ruminants as well [95]. Bovine staphylococcal mastitis, the most thoroughly investigated staphylococcal infection among ruminants, stems from such asymptomatically colonizing bacteria on the animals or the milker’s hands, or from other infected animals, which are introduced via the teat canal [120]. Among the many *S. aureus* clonal complexes (CCs) that have been associated with bovine mastitis, CC97 is the most frequent [93]. Other widespread CCs causing bovine mastitis are CC1, CC5, CC8 (which are common causes of human infections), and CC398 (which is a common source of infection in pigs) [121]. The latter were probably transferred from humans and pigs, respectively, by relatively recent transmission events [122,123,124]. CC97 is predicted by genome analysis and according to in vivo studies to have pronounced virulence [125,126], but other lineages, notably CC479, have also been associated with pronounced virulence potential [127]. Recent studies on CC97 and ST59, the major mastitis clone in Asia, have indicated that there were a series of host switching events between humans and cattle [90,91,92]. There is evidence that these were accompanied by host adaptation, for example increasing lactose utilization in cattle after human-to-cattle jumps or human innate immune evasion capacity after cattle-to-human jumps [92].

*S. aureus* virulence determinants that have a potential impact on mastitis are manifold and for the most part reflect the mechanisms described in human infections. However, evidence is generally obtained only from gene content, in vitro experiments, and correlative analyses—only very rarely from direct analysis in mastitis models.

As in humans, adhesion to tissues is facilitated by members of the MSCRAMM family [40]. Some discrepancies with human infection have been found, including fibrinogen-independent adhesion to bovine mammary epithelial cells via ClfA [128]. Probably owing to pronounced functional redundancy among MSCRAMMs, there is great variation in the MSCRAMM repertoire in *S. aureus* isolates obtained from mastitis [121]. Given that mastitis is a chronic infection, biofilm formation is widely believed to contribute to it [129,130], but there is no direct evidence supporting that notion. Rather, this assumption is based on in vitro analysis of mastitis isolates, many of which have the *ica* genes coding for the exopolysaccharide PIA/PNAG or the gene encoding biofilm-associated protein (Bap) [131], which is carried by a putative composite transposon inserted in the bovine *S. aureus* pathogenicity island SaPIbov2. However, the presence of *bap* in *S. aureus* isolates of animals, including those of bovine or human origin, appears to be very low [132]. Superantigenic toxin and enterotoxin gene content is highly variable in mastitis isolates, as these factors are mostly encoded on MGEs [121]. The presence of specific enterotoxin genes has been associated with acute clinical mastitis [133]. In a mastitis model in dairy cattle, a derivative of the bovine strain RF122, in which eight superantigen genes were deleted, more rarely caused clinical mastitis as compared to wild-type RF122 [134]. While the *hla* gene for alpha-toxin was deleted in both strains in this study to limit overshadowing alpha-toxin-related virulence, these results indicate a role for superantigens in clinical mastitis. As for alpha-toxin itself, an *hla* mutant of a mastitis strain showed reduced mortality after intra-mammary injection in a mouse mastitis model, but not reduced survival in the mouse mammary gland [135]. Leukocidins are deemed particularly important to establish intramammary infection by providing resistance to invading neutrophils [136]. Some strains from cattle and other animals contain a LukF variant, LukF-P83 (LukF’), as part of the leukocidin LukF’M, which is associated with main CCs causing mastitis in cattle [46]. Finally, PSMs have been reported to decrease the production of some interleukins, especially IL-32, in bovine mammary epithelial cells [137], which somewhat contrasts the generally pro-inflammatory effects of PSMs that is mediated by stimulation of the formyl peptide receptor 2 (FPR2) [138]. Interestingly, the bovine origin RF122 strain revealed very limited production of PSMs other than the delta-toxin, reflecting the situation in some laboratory human strains of *S. aureus* such as 8325-4 [137].

The differences in host-specific virulence capacity and mechanisms that are associated with *S. aureus* from specific animals, or humans, are believed to be mainly due to differences in MGEs and the factors encoded on them. According to a compilation by Haag et al. [8], in addition to what was mentioned above regarding *bap*, ruminant-specific MGEs and associated virulence antibiotic resistance factors comprise the following: (i) enterotoxins encoded on the bovine pathogenicity island SaPIbov (*sec*-bovine, *sel*, and *tsst*-1) [139,140] and an enterotoxin cluster (*seg*, *sei*, *sem*, *sen*, and *seo*) [140,141]; (ii) the staphylococcal superantigen-like genes *ssl07* and *ssl08* [142], the gene encoding von Willebrand factor binding protein (vWbp) on SaPIbov4 [143], a gene coding for an LPXTG surface protein on a non-*mec* staphylococcal cassette chromosome (SCC*mec*) element [122], and the *mecA* homolog, *mecC*, on SCC-*mecC* [144]. Contrastingly, the *phiSa3* (beta-hemolysin converting phage) that encodes several immune evasion factors, such as the chemotaxis inhibitory protein of *S. aureus* (CHIPS) and staphylococcal complement inhibitor (SCIN), in addition to several enterotoxin genes, appears to be specific to humans and is absent in other animals [92].

Bovine mastitis is caused by non-*S. aureus* species in about 5–12% of cases, as studies from different countries indicate [145,146,147]. Intramammary infections (IMIs) by non-*S. aureus* staphylococci seem to be increasing in relative frequency as compared to those due to *S. aureus*, potentially due to *S. aureus*-focused control measures. Several studies revealed CoNS/non-*S. aureus* staphylococci as the predominant causes of IMIs/subclinical mastitis in cows [92,148]. The predominant non-*S. aureus* staphylococcal species involved with IMIs is *S. chromogenes* [149,150]. This species appears to be adapted to ruminant hosts [151,152] and shows a higher virulence potential than other non-*S. aureus* species, as indicated by a higher inflammatory capacity and an increased duration of IMIs [153,154].

Staphylococci are also the most frequent pathogens associated with mastitis in sheep and goats [96]. In these animals, like in cows, *S. aureus* is the most common cause of clinical mastitis, and non-*S. aureus* staphylococci typically cause subclinical mastitis [96,97]. CoNS are the most frequent cause of subclinical mastitis in small ruminants, making up >70% of infectious isolates obtained from such infections in sheep and goats [100]. *S. caprae* is the predominant infectious CoNS in goats, as it appears to exhibit pronounced host specificity [100]. According to its genome, *S. caprae* has a virulence potential comparable to that of other CoNS such as *S. epidermidis* [155]. Immune responses due to CoNS mammary infection were similar among goats, sheep, and cows, but showed higher leukocyte numbers in goats [156]. After experimental inoculation with *S. chromogenes*, goats showed increased signs of inflammation [98]. Finally, Morel’s disease is a sort of lymphadenitis that is restricted to sheep and goats and caused by a host-specific microaerophilic subspecies of *S. aureus*, *S. aureus* subsp. *anaerobius* [99,157]. The role of specific staphylococcal virulence mechanisms in small ruminant diseases remains poorly explored.

Antimicrobial resistance plays a considerable role in ruminant infections. LA-MRSA has already been discussed above. With regard to ruminant-infecting staphylococci, the average values of MRSA prevalence in bulk milk from dairy cows and in individual milk samples from more than one farm are ~2.9% and ~4.5%, respectively [158]. However, in specific cases, values of up to ~50% were observed, and considerable geographic variation exists. In Europe, CC398 is the most prominent LA-MRSA lineage in dairy herds [158]. In a recent study performed in several countries, prevalence of penicillin and erythromycin resistance in *S. aureus* isolated from cases of clinical mastitis was about 20%, while methicillin resistance was sporadic [159]. In another study analyzing Belgian and Norwegian isolates from milk samples, including non-*S. aureus* staphylococci, resistance to trimethoprim-sulfonamide was frequent in *S. epidermidis* and *S. haemolyticus*, while *mecA* was harbored in 10 out of 64 isolates from Belgium but was absent from isolates obtained in Norway [160]. These studies also indicated that frequency of the *mecC mecA*-homolog, which was first described in 2011 [144], appears to still be very low in cattle, and similar findings were achieved in goats [161].

**Table 2 ijms-24-14587-t002:** Summary of virulence factors from pathogenic staphylococcal species with demonstrated functions in animal infections.

Staphylococcal Species	Virulence Factor	Description	Reference(s)
*S. aureus*	PNAG/PIA	Biofilm formation	[162]
	FnBP	Binding of fibronectin	[163]
	von Willebrand factor-binding protein (vWbp)	Plasma coagulation	[143]
	Enterotoxin gene cluster	Superantigens	[134]
	Alpha-toxin	Pore-forming toxin	[135]
	LukF’M	Bicomponent leukocidin	[46]
	Phenol-soluble modulins (PSMs)	Cytolytic/proinflammatory peptide toxins	[137]
	ScpA	Thiol protease	[164]
	SAAV_0062 and SAAV_0064	Unknown, allow growth at 42 °C	[164]
*S. pseudintermedius*	PSMs (Delta-toxin and PSMepsilon)	Cytolytic/proinflammatory peptide toxins	[165]
	SIET, ExpA (EXI), ExpB	Exfoliative toxin	[166,167,168]
	LukI	Bicomponent leukocidin	[48]
	SEC_CANINE_	Superantigen	[169]
	SpsP, SpsQ	Immune evasion (binding of IgG Fc, altering B cell function)	[170,171]
	SpsD, SpsO	Cell wall-anchored proteins involved in adherence	[171]
	NucB/AdsA	Nuclease/adenosine synthase	[172]
*S. hyicus*	SHETA, SHETB, ExhA, EXhB, ExhD	Exfoliative toxins	[101,173,174,175]
	Protein A homolog	Immune evasion (binding of IgG Fc, altering B cell function)	[176]
	Lipase	Cleaves triglyceride lipids	[177,178]
*S. chromogenes*	SCET, ExhB	Exfoliative toxins	[103,179]
*S. felis*	PSMs (delta toxin, PSMbeta 1–3)	Cytolytic/proinflammatory peptide toxins	[180]
*S. xylosus*	PSMs (PSM⍺, PSMβ1)	Cytolytic/proinflammatory peptide toxins	[181]
	SxsA	Cell wall-anchored protein involved in adherence	[182]

### 4.2. Staphylococcal Pathogens in Dogs and Cats

Skin disease (pyoderma) and infections of the external ear canal (otitis externa) and the urogenital tract are main reasons for seeking veterinary attention for cats and dogs [6,7,183,184,185,186]. Pyoderma is seen as a spectrum of diseases and is the most frequent infection observed in household pets caused by staphylococci. The least invasive form (superficial bacterial folliculitis) is characterized by pustules, alopecia, erythema, crusts, scaling, and pruritus that may proceed to more deep-seated and painful forms (furunculosis and cellulitis) [6,187,188]. While pyoderma skin infections are not life-threatening, they can have a profound impact on well-being and health. Dogs with underlying food and environmental allergen sensitivities as well as pre-existing inflammatory dermatological conditions, such as atopic dermatitis (AD) [189,190,191,192,193], are more prone to pyoderma [189] and ear-canal infections [194]. It is thought that AD-induced scratching leads to mechanical damage to the skin barrier and concomitant transfer of staphylococci to these inflamed sites from licking or grooming [195], facilitating bacterial penetration into the upper skin layers, which can lead to secondary infections.

Molecular typing methods have redefined the taxonomy and host associations of *S. intermedius* group members. In two pivotal studies, all isolates recovered from dogs, cats, and humans were identified as *S. pseudintermedius*; thus, *S. pseudintermedius* (not *S. intermedius*) was revealed as the common causative agent of canine pyoderma [196,197]. *S. pseudintermedius* is a primary commensal of canines [6,198] and is also detected in other non-canine species including horses and cats [199]. However, the colonization rates in these other animal species tends to be low [188,200]. For instance, cats show a ~6.5 fold-lower colonization rate compared to dogs [188,200]. As *S. pseudintermedius* is frequently isolated from canine skin-, urine-, and ear canal-infections [6,105,106,110,183,201,202], there is general agreement in the veterinary field that *S. pseudintermedius* is the major etiological agent of canines, and despite the recent emergence of *S. coagulans* [12,107,203], it has received most attention as a canine pathogen. In contrast, the coagulase negative staphylococcal species, *S. felis*, appears to be the common cat commensal and is more frequently isolated from sick animals [184].

Over the last two decades, the global emergence of multidrug resistance among methicillin-resistant *S. pseudintermedius* (MRSP) isolates from canine infections has created a serious challenge to the veterinary sector [6]. MRSP infections complicate treatment procedures and necessitate multiple drugs to clear infections, thereby prolonging disease resolution and encouraging the development of more antimicrobial resistance [6,204,205]. Given the high frequency of multi-drug-resistant, disease-causing staphylococcal pathogens that already exist in humans (e.g., *S. aureus*, *S. epidermidis*, and *S. haemolyticus* [18,27]), zoonotic transmission of MRSP to humans poses a further public health risk, especially for pet owners and veterinary staff [206,207]. In humans, MRSP-associated soft tissue infections, occurring through dog bites from colonized or diseased animals [208], are becoming increasingly common. Moreover, MRSP-invasive bloodstream infections have been described but are restricted to elderly individuals in nosocomial settings or those with pre-disposed health conditions, such as diabetes [209,210,211]. On the other hand, the incidence of multidrug resistance in *S. felis* is rare [112,212], and only one case of cat-to-human transmission has been reported to date [213].

Like its pathogenic human cousin *S. aureus*, *S. pseudintermedius* possesses an active Agr quorum-sensing system [214]. *S. pseudintermedius* Agr has been described in detail [214,215,216]. Interestingly, there are four AIP alleles in *S. pseudintermedius* [216] that each contain a serine in place of the conserved AIP cysteine residue, resulting in the formation of a cyclic lactone rather than the thiolactone ring that is typically present in other staphylococcal AIPs [214,215,216]. Activation of the *S. pseudintermedius* Agr system results in the transcription of some toxin genes [214], but it is currently unknown if Agr has a crucial role in the development of *S. pseudintermedius* skin infections or other diseases, as has been established for the major human pathogens *S. aureus* [217] and *S. epidermidis* [218]. It is clear from whole genome sequencing [219] that *S. pseudintermedius* harbors a multitude of other putative virulence genes, including those that are involved in adherence, biofilm formation, and immune evasion, and a plethora genes encoding a diverse array of toxins [11].

One such family of toxins are PSMs; it was previously shown that that a delta-toxin gene homolog (*hld*) is present in *S. pseudintermedius* [214]. More recently, it has been found that clinical *S. pseudintermedius* isolates produce one of two delta-toxin variants and an additional PSM, PSMepsilon [165]. Both delta-toxin variants and PSMepsilon are cytolytic, similar to the alpha-type PSMs of *S. aureus* [220] and *S. epidermidis* [221]. However, in the context of canine skin infections, the PSMs may have an additional role. A breakthrough study by Nakamura et al. [222] first highlighted a central role of the mast cell-degranulating properties of *S. aureus* delta-toxin for the development of AD. However, a later study implied that mast cell degranulation was a general feature of PSMs [223]. *S. pseudintermedius* PSMs may exacerbate AD in canines, but this remains to be investigated.

In addition to PSMs, canine strains of *S. pseudintermedius* produce a number of other toxins that could potentiate pyoderma infections, including three different exfoliative toxins, SIET [166], ExpA (formally known as EXI) [167], and ExpB [168]. Dogs subcutaneously injected with purified SIET and ExpA developed clinical signs such as erythema, exfoliation, and crusting. Only ExpB has been shown to target the canine form of the cadherin transmembrane protein, Dsg-1 [168]. Other toxin genes found in the *S. pseudintermedius* genome include a bicomponent leukocidin LukI [224], which targets polymorphonuclear white blood cells from canine origin [48], a beta-hemolysin [225] and enterotoxin SEC_CANINE_, which acts as a superantigen [169]. Further enterotoxin genes have been identified by analytical PCR and by whole genome sequencing in *S. pseudintermedius* [226,227].

Otherwise, *S. pseudintermedius* harbors a repertoire of virulence genes similar to *S. aureus* such as the biofilm-forming *ica* genes [228] and two protein A orthologs (SpsP and SpsQ) [229], whose IgG-binding activities were only recently characterized [170,230]. In addition to SpsP and SpsQ, *S. pseudintermedius* is predicted to encode 16 other putative cell-wall-anchored surface proteins [219,229]. The introduction of two cell-wall-anchored genes (*spsD* and *spsO*) into a heterologous host, *Lactococcus lactis*, resulted in increased adherence to canine corneocytes ex vivo [171].

It has also been shown that, similar to *S. aureus* [231,232], *S. pseudintermedius* relies on adenosine synthase A (AdsA) for abscess formation in a systemic model of bloodstream infection in mice [172]. The underlying mechanism consists of a two-step process in which the nuclease (NucB)-dependent breakdown of host DNA promotes the release of deoxyadenosine monophosphate (dAMP). AdsA converts dAMP into a cytotoxic derivative that ultimately kills macrophages and therefore impairs the host’s ability to control infection.

In contrast, little has been done to characterize *S. felis* or its virulence factors. Information from whole genome sequencing reveals the presence of genes associated with adhesion, immune evasion, biofilm formation, and toxin and proteolytic enzymes [212]. However, three PSMbeta peptides and a delta-toxin were recently identified in an *S. felis* isolate [180]. As in other staphylococci, only the delta-toxin, not the PSMbeta peptides, showed pronounced cytolytic activity [45,180,220].

### 4.3. Staphylococcal Infections in Swine

*S. hyicus* and *S. chromogenes* form a major part of the normal microbiota of pigs as well as poultry and cattle [233,234], and both species can be isolated from swine exudative epidermitis (EE) [101,102,179]. EE is a major skin disease of pigs. Recently weaned piglets are the most vulnerable population. The disease is best characterized by the enhanced production of a greasy exudate, giving rise to its common name “greasy pig disease”. Clinical symptoms, initially most prevalent surrounding the pinnae and in the axillary, inguinal, and abdominal areas, can eventually spread systematically, leading to more serious outcomes. In those cases, exfoliation, pyoderma, erythema, and the development of crusts [235] may manifest across the entire body, turning the skin brown within 24–48 h [236,237]. Piglets with EE often suffer from dehydration and malnourishment, and thus, high mortality rates are observed [101,173]. Compared to *S. hyicus*, the association of *S. chromogenes* with EE is less frequent [103], which highlights *S. hyicus* as the principal agent underlying EE disease [101,102].

The increase in demand for pork products has resulted in concentrated industrial-scale pig-farming practices alongside the widespread use of antibiotics in feed. Surprisingly, there is little literature describing drug resistance patterns in *S. hyicus* and *S. chromogenes.* However, reports from Brazil [238] and Denmark [239] indicate that antimicrobial resistance patterns of *S. hyicus* isolates change over time and are tied closely to the antimicrobials that are administered. Another study evaluated antimicrobial sensitivities in *S. hyicus* isolates from pigs with EE from 30 herds in Ontario, Canada. Here, it was demonstrated that resistance to one or more antimicrobials was detected in 99.3% (142/143) of *S. hyicus* isolates, with resistance to beta-lactam antibiotics being the most prevalent [240]. More worrisome was that, in the same study, 40.6% (58/143) of *S. hyicus* isolates demonstrated resistance to five or more antimicrobials. There is even less data regarding *S. chromogenes*, but according to the few reports available, it is not uncommon for *S. chromogenes* to be resistant to more than one drug [240,241].

The clinical manifestations of EE are similar to those observed in human staphylococcal scalded skin syndrome (SSSS), a disease afflicting neonates and young children [242,243]. SSSS is a systemic infection involving fever, malaise, substantial degrees of blister formation, and exfoliation of superficial skin over large parts of the body [244,245] and is seen as the advanced stage of bullous impetigo, a local form of SSSS [246]. In SSSS, pathogenesis is mediated by *S. aureus* strains that produce exfoliating toxins. As the toxins move into the *stratum granulosum* of the epidermis, followed by the exfoliative toxin-mediated cleavage of Dsg-1 [55,56], these intercellular interactions between stratified squamous epithelia are disrupted [246] causing intraepidermal desquamation and a subsequent formation of blisters [247,248]. *S. aureus* itself is not found in the exfoliation sites or from cultures of the bullae, indicating that the exfoliative toxins can spread systemically from the primary site to other parts of the body [246].

EE pathogenesis is also mediated by exfoliative toxins as demonstrated by a strong association between exfoliative toxin expression in *S. hyicus* and EE in pigs [101,174,249,250]. Six different exfoliative toxin genes have been discovered to date: a chromosomally-encoded SHETA [173], a plasmid-encoded SHETB [251,252]), and four Exh variants (ExhA, ExhB, ExhC, and ExhD) [101,253], whose genes are located on a pathogenicity island or a prophage-related element [254,255]. While all recombinant Exh proteins have the capacity to specifically cleave swine Dsg-1 in vitro [175,256], this has not yet been demonstrated for SHETA and SHETB. However, pathogenesis is observed after subcutaneous injection with purified toxins or by comparing toxin- and non-toxin-producing *S. hyicus* strains in piglets [101,173,174,175].

Other virulence determinants of *S. hyicus* include delta-toxin [255], an IgG-binding protein A homolog [176], and a lipase [177,178]. Although a role in *S. hyicus* pathogenesis has not been demonstrated, expression of *S. hyicus* lipase in a heterologous host (*S. aureus*) promotes biofilm formation and an invasion of keratinocytes [257]. Other potential interesting genes that could contribute to *S. hyicus* pathogenesis are located in two genomic regions present in toxigenic strains [254].

In *S. chromogenes*, putative virulence genes identified from whole genome sequencing include those associated with biofilm formation, attachment, and immune evasion [258,259]. Additionally, two exfoliative toxins, SCET [179] and ExhB [103] have been described. As the only studies conducted with SCET [179] and ExhB [103] were performed in pigs without adequate controls, a contribution towards EE is difficult to endorse. However, in the latter study, subcutaneous injection behind the ear with an ExhB-expressing *S. chromogenes* strain caused local exfoliation and at a distal site (the hoof), which is reminiscent of the clinical symptoms featured in SSSS disease. No further research has been conducted on these toxins since their discoveries.

### 4.4. Staphylococcal Infections in Chicken

An increased demand for poultry meat over the past several decades has forced alterations in the way birds are reared for production, such as the genetic selection of birds that grow faster. In the past, it would take 120 days for a broiler chicken to reach 1.5 kg in body weight, whereas nowadays the same weight can be reached in a quarter of that time [260]. This rapid and excessive accumulation of body weight in short periods has a profound impact on the mobility of the birds [260]. Moreover, these practices are linked with abnormal skeletal development in the leg bone tissue, resulting in increased bone deformities and leg trauma, lameness, and pain [260,261,262]. Poor leg health and lameness can occur from noninfectious and infectious origins [263]. In the latter, bacterial chondronecrosis with osteomyelitis (BCO) is a leading cause of lameness [264] and defined as a septic necrosis of the skeletal system. In BCO, bacteria may find entry into the bones through microfractures and clefts, caused by the mechanical stress of walking, which eventually causes micro- and macroscopic lesions [264]. BCO lesions can be detected in 28% of the mortalities and culls [265]. Furthermore, bacterial endocarditis and bacteremia are serious systemic diseases that appear sporadically in poultry farms, but their prevalence is not well documented in the literature. However, it is estimated that bacteremia accounts for up to 62% mortality in broiler breeders [266], while mortality rates due to endocarditis outbreaks in flocks are between 29 and 36% [266,267,268]. Lastly, a dermatological disease called Focal Ulcerative Dermatitis Syndrome (FUDS) has recently been reported in cage-free laying flocks [269]. FUDS is characterized by the development of lesions on the dorsal regions of the birds, next to the sebaceous uropygial gland [269]. Hens with clinical signs of FUDS experience significant decreases in egg production and loss of life [269]. Altogether, these diseases result in substantial economic losses in the poultry industry.

Staphylococci are often found in diseased birds from poultry farm environments. In particular, *S. aureus* and *S. hyicus* can be isolated from osteomyelitis and systemic infections, but other staphylococcal species are also found [113]. One of these is the coagulase-variable *S. agnetis*, which historically has been associated with clinical and subclinical cases of bovine mastitis [94]. *S. agnetis* has also been detected in BCO lesions [114] and cases of septicemia in broiler breeder flocks [115]. *S. aureus* and *S. agnetis* have been described as potential causative agents of FUDS [269]. Antimicrobial resistance in poultry isolates of *S. agnetis* and *S. hyicus* has not been well characterized. Multidrug resistance was reportedly not prevalent in *S. agnetis* isolates from a commercial laying hen operation with a history of FUDS in the mid-west United States [269], while a report from Denmark noted a high prevalence of multidrug resistance in *S. hyicus* poultry isolates [270].

In contrast to cattle where staphylococcal infections are mostly associated with the mammary glands, *S. agnetis* infections in poultry include those of the bone, blood, and organs. Little is known about what drives *S. agnetis* infections. Whole genome analyses of *S. agnetis* isolates from cattle [271] and poultry [272] indicate the presence of genes commonly involved in staphylococcal pathogenesis, such as toxins (a PSMbeta peptide, an exfoliative toxin A homolog, multiple superantigens, and a beta-hemolysin), genes for cellular adherence (seven fibronectin-binding proteins, elastin- and fibrinogen-binding proteins, and a collagen adhesin), biofilm formation, immune evasion, and an Agr quorum-sensing system. To explain the recent epidemic success of *S. agnetis* in poultry, Shwani et al. compared the genome sequences of cattle and poultry isolates of *S. agnetis* [273], but no distinguishing genes to explain recent trends or host tropism were found. In contrast, some chicken-adapted *S. aureus* isolates appear to have undergone significant genetic changes. For instance, one study discovered that genetic recombination events happened in specific poultry isolates of the CC5 lineage, which caused the loss of genes involved in human pathogenesis but also allowed several host adaptations, including the improved ability to grow at the core body temperature of chickens (42 °C), the inhibition of neutrophil activation and chemotaxis via a thiol protease, staphopain A, and the increased lysis of chicken erythrocytes [164].

### 4.5. Staphylococcal Infections in Mice

Bacterial skin infections are a common occurrence in laboratory mice, especially in mice of the C57BL/6J background, which have defects in immune function resulting from genetic manipulation [274,275]. The coagulase-negative *S. xylosus* is the dominant commensal of laboratory mice [276,277] and is commonly thought to be the major contributor to murine skin infections, as evidenced by its frequent isolation from skin lesions of mice suffering from AD-like symptoms [278,279,280,281,282].

The observation that *S. xylosus* could be detected in murine AD-like lesions is especially interesting because of the association of *S. aureus* with AD in humans [283,284] where it was shown to be dependent on *S. aureus* PSMs triggering mast cell degranulation [222,223]. While the overall capacity of *S. xylosus* to promote AD was shown to be significantly weaker than that of *S. aureus*, two main PSMs were identified in *S. xylosus*, PSMalpha and PSMbeta1, which may trigger inflammatory responses responsible for AD-like symptoms in mice [181]. The development of such responses likely requires a predisposed host [281,282]. Of note, the mast cell-degranulating properties of the *S. xylosus* PSMs correlated with their cytolytic capacities, emphasizing the notion that this pathogenic feature is not receptor-mediated. However, PSMs may also interact with other cell types, raising the question of whether the progression of AD could also be attributed to PSM-dependent activation of FPR2, e.g., in keratinocytes. Lebtig et al. recently showed that FPR2 inhibition or the absence of the receptor in FPR2^−/−^ mice prevented the release of key inflammatory cytokines in a mouse model of AD [285].

Genomic analysis of an *S. xylosus* isolate collected from mouse feces did not reveal the presence of toxin genes (other than PSMs), in accordance with other CoNS [286]. Interestingly, *S. xylosus* can be recovered at high frequencies two weeks after topical application onto the ears of C57BL/6 mice, which is in contrast to other staphylococcal species, such as *S. aureus* [287], suggesting that *S. xylosus* may express determinants that better allow it to survive on or adhere to murine skin. Two genome-encoded biofilm genes, *bap* and *sxsA* (a novel virulence factor), were recently identified in *S. xylosus* isolates from raw fermented sausages [182,288]. It was shown that only SxsA had a major importance in the biofilm formation of *S. xylosus* in vitro.

Lastly, *S. aureus* should not be discounted as a mouse pathogen, even though a recent microbiome analysis showed that it was not present in mouse colonies from distinct facilities across Europe [276]. The *S. aureus* strains that were found to circulate in mouse colonies appear to have adapted to the mouse host [116,289]. Interestingly, mouse-adapted *S. aureus* isolates lack the beta-hemolysin-converting phages that harbor genes for superantigens and human-specific immune evasion factors, a genomic feature that also exists in other animal isolates [290].

## 5. Conclusions and Outlook

Staphylococcal infections in animals not only are of great importance for animal welfare but also affect the agricultural economy to an increasingly considerable extent, owing to the extensive and in part still uncontrolled use of antibiotics in that sector. Human health is strongly impacted by staphylococcal animal infections, due to at least temporary host jumps and resulting infections in humans and because antibiotic overuse in livestock represents a reservoir for the spread of antibiotic resistance genes.

Some staphylococcal strains have pronounced host specificity, while in others, such as *S. aureus*, host adaptation has been observed and linked to specific genetic determinants. To understand how these species and strains cause infections in animals, dedicated pathogenesis research is needed, as findings obtained in humans on virulence factors and mechanisms may have only limited bearing for animal infections. Such research is imperative for the development of anti-virulence drugs, which may represent one possible avenue to deal with the problem of increasing antimicrobial resistance that is especially pronounced in livestock.

However, except for some more in-depth research into the molecular underpinnings of the pathogenesis of *S. aureus*-mediated bovine mastitis, our understanding of virulence mechanisms underlying staphylococcal animal infections is still very limited. Challenges that will be key to change this situation comprise first and foremost the development of genetic tools that can manipulate non-*S. aureus* staphylococci of relevance for animal infections, which has proved difficult in many cases, and the setup of infection models in the animals under consideration.

## Figures and Tables

**Table 1 ijms-24-14587-t001:** Summary of most frequent animal diseases and associated staphylococcal pathogens.

Animal Species	Disease Type	Associated *Staphylococcus* Species	Reference(s)
Cattle	Mastitis	*S. aureus*	[90,91,92,93]
	Subclinical mastitis	*S. agentis*	[94]
		*S. chromogenes*, *S. simulans*, *S. haemolyticus*, *S. xylosus*, *S. epidermidis*	[95]
Sheep	Mastitis	*S. aureus*	[96,97]
		*S. chromogenes*	[98]
	Lymphadenitis	*S. aureus* subsp. *anaerobius*	[99]
Goats	Mastitis	*S. aureus*	
		*S. caprae*	[100]
	Lymphadenitis	*S. aureus* subsp. *anaerobius*	[99]
		*S. caprae*	
Pigs	Exudative dermatitis	*S. hyicus*	[101,102]
		*S. chromogenes*	[103]
Horses	Skin and soft tissue infection, bacteremia, septic arthritis, osteomyelitis, implant- and catheter-related infections, metritis, omphalitis, pneumonia	*S. aureus*	[104]
Dog	Atopic dermatitis, pyoderma	*S. pseudintermedius*	[105,106]
		*S. coagulans*	[12,107]
	Otitis externa	*S. pseudintermedius*	[108,109]
		*S. coagulans*	[12,107]
	Urinary tract infections	*S. pseudintermedius*	[110]
Cat	Pyoderma	*S. aureus*	[111]
		*S. felis*	[111]
	Urinary tract infections	*S. felis*	[112]
Avian	Chondronecrosis with osteomyelitis	*S. aureus*	[113]
		*S. agnetis*	[114]
	Systemic infections	*S. aureus*	[113]
		*S. hyicus*	[113]
		*S. agnetis*	[115]
	Focal Ulcerative Dermatitis Syndrome	*S. aureus*	[116]
		*S. agnetis*	[116]

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
