# Peer review of "Virulence Mechanisms of Staphylococcal Animal Pathogens"

_ijms, 2023, doi:10.3390/ijms241914587_

Round 1

Reviewer 1 Report

This review paper by Cheung and Otto describes overview of virulence mechanisms of staphylococcal species causing infections in animals, occasionally contrasting to humans. This is well written and gives readers latest findings on the topic. This reviewer has following minor points to improve this manuscript.

1. The title of this manuscript should be reconsidered. Please check whether the phrase "staphylococcal animal pathogens" represents the content of manuscript appropriately.

2. line 116-119: Five different leukocidins are listed. However, Table 2 includes Luk FM for S. aureus, but it is not mentioned in this paragraph. Luk FM should be explained. In past literatures, "LukM" has been commonly described, which should be also mentioned.

3. line 194-195: "resistance to methicillin is the most important antibiotic resistance determinant" is a strange sentence. Resistance determinant is mecA or SCCmec. Please rephrase. The name of MRSA, i.e., methicillin-resistant S. aureus has been commonly described by researchers, though methicillin is not used at present, as historically established name. MR is generally regarded as representing beta-lactam resistance. However, some readers may not be familiar with it. Authors should add some more explanations regarding this. 

4. line 234: "CC1, CC5, CC8 (a CC that is also a common cause of human infections)" This sentence may be misunderstood that only CC8 is a common cause in humans, although CC1, CC5, CC8 are all as such CC. Please rephrase.

5. line 262 "MGEs", line 301 "IMI" should be shown with full spelling of words if they appear first in the text.

6. Section 4.4. In this section, S. fleurettii is not mentioned. This species is considered to be an origin of mecA for MRSA. Does this species not cause infectious disease in chicken or humans, and does zoonotic  transmission occur commonly? Consider whether it is necessary to add this species in this section or not.

7. Overall, authors covered most staphylococcal species that cause infections in animals. However, there is no description about Mammaliicoccus, that includes former S. sciuri. If authors do not include this group in this review, it is better to write in early part, in the paragraph of CoNS. Otherwise, Mammaliicoccus should be also mentioned in text, if necessary. 

Author Response

This review paper by Cheung and Otto describes overview of virulence mechanisms of staphylococcal species causing infections in animals, occasionally contrasting to humans. This is well written and gives readers latest findings on the topic. This reviewer has following minor points to improve this manuscript.

We much appreciate the reviewer’s suggestions for improving the manuscript.

  1. The title of this manuscript should be reconsidered. Please check whether the phrase "staphylococcal animal pathogens" represents the content of manuscript appropriately.

We thought of several alternative titles but we believe that this short and concise title accurately reflects the content of this review. However, if our title does not appease, we welcome suggestions by the editor. 

  1. line 116-119: Five different leukocidins are listed. However, Table 2 includes Luk FM for S. aureus, but it is not mentioned in this paragraph. Luk FM should be explained. In past literatures, “LukM” has been commonly described, which should be also mentioned.

Thank you for highlighting this. We have updated this section and indicate Leukocidin ED (LukED), Panton–Valentine Leukocidin (PVL or LukSF–PV), gamma-hemolysins AB and CB (HlgAB and HlgCB), and Leukocidin AB (LukAB; also known as LukGH) are associated with human infections, and introduce LukF’M, LukI, and LukPQ as being associated with animal infections. We amended nomenclature of all leukocidins to be consistent throughout the text.

  1. line 194-195: "resistance to methicillin is the most important antibiotic resistance determinant" is a strange sentence. Resistance determinant is mecA or SCCmec. Please rephrase. The name of MRSA, i.e., methicillin-resistant S. aureus has been commonly described by researchers, though methicillin is not used at present, as historically established name. MR is generally regarded as representing beta-lactam resistance. However, some readers may not be familiar with it. Authors should add some more explanations regarding this. 

We acknowledge this suggestion by adding several sentences explaining the basic mechanism underlying methicillin resistance afforded by acquisition of mecA in SCCmec.

  1. line 234: "CC1, CC5, CC8 (a CC that is also a common cause of human infections)" This sentence may be misunderstood that only CC8 is a common cause in humans, although CC1, CC5, CC8 are all as such CC. Please rephrase.

This has been changed to “Other widespread CCs causing bovine mastitis are CC1, CC5, CC8 (which are common causes of human infections)…

  1. line 262 "MGEs", line 301 "IMI" should be shown with full spelling of words if they appear first in the text.

MGE is now introduced fourth paragraph in Section 3. IMI is introduced fully in the 6th paragraph of Section 4.1

  1. Section 4.4. In this section, S. fleurettii is not mentioned. This species is considered to be an origin of mecA for MRSA. Does this species not cause infectious disease in chicken or humans, and does zoonotic  transmission occur commonly? Consider whether it is necessary to add this species in this section or not.

A literature search did not reveal that S. fleurettii (now reassigned to the genus Mammaliicoccus with Mammaliicoccus (M.) sciuri as the type species (Madhaiyan et al., 2020) as a major causative agent of disease in chickens. Therefore, we do not feel it is necessary to mention this species, or the genus Mammaliicoccus (see point 7), in our literature review.

  1. Overall, authors covered most staphylococcal species that cause infections in animals. However, there is no description about Mammaliicoccus, that includes former S. sciuri. If authors do not include this group in this review, it is better to write in early part, in the paragraph of CoNS. Otherwise, Mammaliicoccus should be also mentioned in text, if necessary. 

As mentioned in point 6, S. sciuri has been reassigned into the genus Mammaliicoccus. Therefore, we do not feel that discussing M. fleurettii or M. sciuri in relevant to our review.

Reviewer 2 Report

The manuscript is well described and an informative review on virulence mechanisms of various staphylococcal species of animal origin.

Some figures and illustrations should be included for better understanding of readers.

The manuscript is well described and an informative review on virulence mechanisms of various staphylococcal species of animal origin.

In this review, authors made the overview of staphylococcal virulence factors such as leucocidin family and surface proteins (MSCRAMMs), exfoliative toxins and enterotoxins produced by animal staphylococcal pathogens as per data reported previously. They also mentioned about the various staphylococcal infections depending on animal hosts and summarized the virulence factors of pathogenic staphylococcal species with demonstrated infection in specified animal host.

This reviewer suggests following points to improve the quality of this manuscript.

1. This manuscript focuses on Staphylococcus from animal. Therefore, readers may be interested in difference in prevalence, mechanism of action of virulence factors of staphylococcus between animals and humans. A Table or figure/illustration showing staphylococcal virulence factors in human and animals, and any additional information, would be highly informative for readers. In the present form, readers may feel difficulty to read through the content, without any figure or table.    

2. It seems that in this manuscript, authors did not describe all the virulence factors in one virulence factor category. To make a comprehensive review article, virulence factors should be described thoroughly. For example, enterotoxin (SE), authors mentioned probably most, but not all. How many SEs and SE-like genes were identified to date, and which are related to animal disease? Such latest information should be described in text or in table, or supplementary material, with appropriate references.   

Author Response

The manuscript is well described and an informative review on virulence mechanisms of various staphylococcal species of animal origin.

Some figures and illustrations should be included for better understanding of readers.

The manuscript is well described and an informative review on virulence mechanisms of various staphylococcal species of animal origin.

In this review, authors made the overview of staphylococcal virulence factors such as leucocidin family and surface proteins (MSCRAMMs), exfoliative toxins and enterotoxins produced by animal staphylococcal pathogens as per data reported previously. They also mentioned about the various staphylococcal infections depending on animal hosts and summarized the virulence factors of pathogenic staphylococcal species with demonstrated infection in specified animal host.

This reviewer suggests following points to improve the quality of this manuscript.

  1. This manuscript focuses on Staphylococcus from animal. Therefore, readers may be interested in difference in prevalence, mechanism of action of virulence factors of staphylococcus between animals and humans. A Table or figure/illustration showing staphylococcal virulence factors in human and animals, and any additional information, would be highly informative for readers. In the present form, readers may feel difficulty to read through the content, without any figure or table.    

We thank the reviewer for this suggestion but we did not feel such a presentation in table or figure form to give valuable information. With some exceptions that are mentioned in the text, the main difference between animals and humans is the associated strain and the presence/absence of virulence factors, rather than the virulence factors themselves. Section 2 provides a sufficient summary of important staphylococcal virulence factors, which we then use as a platform to describe, in detail, virulence factors that are specifically involved in the pathogenesis of staphylococcal species in animals (Section 4) and which are summarized in Table 2.

  1. It seems that in this manuscript, authors did not describe all the virulence factors in one virulence factor category. To make a comprehensive review article, virulence factors should be described thoroughly. For example, enterotoxin (SE), authors mentioned probably most, but not all. How many SEs and SE-like genes were identified to date, and which are related to animal disease? Such latest information should be described in text or in table, or supplementary material, with appropriate references.   

We specifically set to distinguish this review by organizing it in a fashion according to animal species rather virulence factor. Under this format, we describe the virulence factors that are most important for staphylococcal species for each animal host.

As for the reviewer’s second point, including an extensive overview of superantigens would make our review heavily imbalanced and would therefore require us to systematically evaluate each of the other virulence factors in the same level of detail. We believe that Section 2 provides a sufficient overview of virulence factors that is appropriate for the general audience. However, to guide the audience who wish to learn more about staphylococcal superantigens, we amended the sentence introducing superantigens in the fifth paragraph of Section 2 and added a reference for a detailed review on superantigens.